# Zero-shot performance analysis of large language models in sumrate maximization

**Ali Abir Shuvro**[ID]**, Md. Shahriar Islam Bhuiyan, Faisal Hussain, Md. Sakhawat Hossen**[ID]*

Department of Computer Science and Engineering, Islamic University of Technology, Gazipur, Dhaka, Bangladesh

* sakhawat@iut-dhaka.edu

## Abstract

Large language models have revolutionized the field of natural language processing and are now becoming a one-stop solution to various tasks. In the field of Networking, LLMs can also play a major role when it comes to resource optimization and sharing. While Sumrate maximization has been a crucial factor for resource optimization in the networking domain, the optimal or sub-optimal algorithms it requires can be cumbersome to comprehend and implement. An effective solution is leveraging the generative power of LLMs for such tasks where there is no necessity for prior algorithmic and programming knowledge. A zero-shot analysis of these models is necessary to define the feasibility of using them in such tasks. Using different combinations of total cellular users and total D2D pairs, our empirical results suggest that the maximum average efficiency of these models for sumrate maximization in comparison to state-of-the-art approaches is around 58%, which is obtained using GPT. The experiment also concludes that some variants of the large language models currently in use are not suitable for numerical and structural data without fine-tuning their parameters.

## Introduction

In the fields of artificial intelligence and Natural Language Processing (NLP), large language models (LLM) are crucial for their ability to comprehend tasks and instantly generate desirable outputs. These models, like GPT-3 and its offspring, have shown to be remarkably adept at comprehending and producing text that resembles human speech [1]. They are essential to a variety of applications, including enhancing sentiment analysis and machine translation, as well as enabling chatbots and virtual assistants [2,3]. Automation of processes that traditionally needed a lot of human interaction is made possible by LLMs, which boosts productivity and efficiency in a variety of sectors. By facilitating more natural and pertinent communication between machines and humans, they also support improvements in human-computer interaction.

LLMs like GPT-3, GPT-4, LLaMA, Falcon, and PaLM2 have immensely rejuvenated the domain of NLP and significantly contributed to providing a newer scope of research [1,4,5]. These language models have their own strengths and weaknesses, making some of them make them more useful than others for specific tasks [6,7]. Zero-shot analysis is essential to

**Data availability statement:** https://github.com/ShojebBhuiyan/Dataset-PLOS_Zero_shot_Performance_Analysis_of_LLM-Version-4. All the data are publicly available in this repository.

**Funding:** The author(s) received no specific funding for this work.

**Competing interests:** The authors have declared that no competing interests exist.

assess the strengths and weaknesses of LLMs because it focuses on the generalization skills of a model by evaluating how well it performs on tasks that it was not explicitly trained for. Researchers learn more about the innate comprehension of language, logic, and context of LLM by putting them through different activities. Zero-shot analysis reveals remarkable adaptability as well as any biases or flaws these models may have, helping us to understand better how to use them in different applications and the need for responsible development and deployment in an artificial intelligence environment [8,9].

The zero-shot capabilities of an LLM can be tested on various tasks. Among other tasks where LLM have proved its capabilities, sumrate maximization is yet to be analyzed. Sumrate maximization is a special task where the settings of a network is optimized to obtain maximum throughput. In this manuscript, the task of Sumrate maximization is taken into consideration. Sumrate maximization is a crucial task in the networking domain, and many algorithms have been introduced for its effective calculation and optimization [10,11]. By effectively distributing resources like bandwidth, electricity, or time slots, sumrate maximization improves the overall data transmission rate of a communication system. It is pivotal for enhancing network performance and overall throughput, particularly when several users use a single channel. It is unclear which algorithm would be suitable for a specific scenario, and with the advent of smaller networks courtesy of IoT devices, the selection gets more difficult. LLMs have solved many such problems due to their simplicity of usage. It does not require prior algorithmic knowledge; only knowing the problem itself would suffice [12].

The key contributions of this research are:

1. Performing a sumrate maximization task leveraging LLMs using multiple iterations to obtain optimized prompt.
2. Comparative analysis of the sumrate generated from LLMs with other simulated algorithms.

The rest of the paper is organized as follows. Sect 1: Related work comprises of the related literature and recent advances in the domain. Sect 2: Methodology gives an overview of the workflow of the experiment. Sect 3: Results and discussion provides the details of the experiment and its results. Sect 4: Result analysis analyzes the results from the experiment. Sect 5: Limitations provides insight on the lack of using LLMs for such tasks. Sect 6: Conclusion and future work gives concluding remarks and discusses the possibilities and scope of future research.

## 1 Related work

NLP has taken a large step with the discovery of large language models. Transformers were introduced using a self-attention mechanism for sequence-to-sequence tasks. The encoder-decoder architecture of the Transformers has revolutionized the NLP domain [13]. NLP models like BERT are powerful, with huge upside potential for improved performances in different NLP tasks [14,19]. It uses a specific implementation of pretrained Transformers to provide contextual understanding [15]. Integrating these tasks into large language models has created a lot of prospects for future research activities [16] starting from complex reasoning abilities through chain of thought to even text-to-image diffusion. Large language models have an inherent capability to find patterns from their training through a vast corpus of conversational datasets. With very little or no data on the current problem, LLMs can figure out solutions from their reasoning capabilities [17]. Newer models of LLMs have developed immensely in different fields and LLMs have greatly advanced in reasoning through chain-of-thought prompting and scratchpad reasoning [18]. While there are many avenues where

large language models have proved helpful, there are limitations and challenges as well. For example, multiple instances have shown that LLMs, including GPT models, have proven to provide inaccurate and sometimes even impossible outputs due to their non-deterministic nature, which illustrates their incompetence in reasoning [20–23].

Prompt engineering is key to optimizing performance for tasks like sentiment analysis or language translation. It entails creating purposeful input queries to refine huge language models like GPT or BERT. Recently, large language models have been rapidly used as a tool for zero-shot outputs. While zero-shot chain-of-thoughts prompting can outperform vanilla zero-shots as it has some idea of the facts, zero-shot results are also quite extraordinary due to the reasoning capabilities of LLMs [24]. Modification of instructions to support a particular need has a massive boost in the performances of the large language models [25]. Even though some authors have argued that LLMs are not zero-shot communicators as they fail to understand context, it is seen that LLMs have significantly improved in understanding contexts and providing outputs that are near-perfect with respect to human interpretation [26]. This makes the zero-shot approach suitable for problem-solving many tasks, including resource allocation.

GPT is one of the leading LLMs in recent times. Its realistic outputs have been termed as a new era for NLP. Zero-shot analysis of the performance of GPT has been done in various scenarios [27]. Information extraction using GPT has been done for unannotated texts [28]. Medical text de-identification has been done using GPT, which shows promising results [29]. Similarly, a comparison of the zero-shot results of GPT with other Transformer-based models trained on biomedical tasks has been evaluated to show its prowess on unforeseen data [30]. Lyrics transcription using GPT has also been tested with its zero-shot capabilities [31]. Again, it has been shown that zero-shot performances of GPT, along with other LLMs, have room for development while analyzing financial tasks [32]. These motivate testing the zero-shot capabilities of LLMs in resource allocation problems.

As LLMs have shown promising results in their zero-shot performances, it can be understood that LLMs would be appropriate for resource allocation problems. In specific, D2D communication requires constant allocation strategies. There are many resource allocation techniques in place for D2D communication [33]. Game theoretic and non-game theoretic resource allocation strategies have been discussed, especially in D2D communications [34]. Machine learning approaches have also been used for resource allocation. Reinforcement Learning approaches have created a new pathway for research in the field of resource allocation in heterogeneous cellular networks [35]. Unmanned aerial vehicles have been used to provide an idea of a D2D energy allocation [36]. Also, LLM-based RAG for network optimization shows the diversity of LLM across various platforms and domains [37]. D2D-Enabled 6G resource allocation has been done using Federated Reinforcement Learning [38]. Modification of the Hungarian Algorithm produced near-optimal results for sumrate maximization [39].

## 2 Methodology

The experiment is conducted on the foundation of [39], where the optimal resource allocation approach for sumrate maximization is compared with the zero-shot performance of LLMs. It involves zero-shot prompt engineering, LLM output generation, simulation of an optimal resource allocation algorithm and comparison of the LLM output and algorithm results. The following subsections, in order, are the steps taken by this experiment to achieve the required results.

## 2.1 Prompt creation

First, a prompt was crafted which describes the properties of a base station, cellular user equipment (CUE) and device-to-device (D2D) pairs in a 2D grid. The properties which are necessary for sumrate and interference calculations are given in Table 1.

The goal was to input CUE and D2D pair property values in a JSON object and return an assignment matrix that allocates cellular user equipment and D2D pairs in sharing resource blocks (RB) that resulted in the maximum sumrate for the network. We adopted a zero-shot approach where the LLM was free to choose any algorithm.

In order to produce the final prompt, several iterations were made for improvements so that the LLM could comprehend the task. In the first iteration, the problem statement and the input and output JSON format were provided with the help of TypeScript interfaces. In the second iteration, the TypeScript interfaces were dropped and the parameters were mentioned explicitly with default base station values. Further conditions were provided to help with allocation such as only one cellular UE can be paired with one d2d pair. The second iteration contained a special instruction that the LLM may choose not to assign some CUE and D2D pairs in an RB if it resulted in a greater network sumrate. The third iteration contained more information about the network. We added information about the communication Carrier Frequency (1.7 Ghz) and a special instruction that the number of CUEs is far greater than the number of D2D pairs. In the final iteration, a final condition was added that all the cell objects and the receiver devices in the d2d pair must obtain their target SNR. The LLM was explicitly instructed to respond only in a JSON object containing the assignment matrix in all the iterations. The final prompt that was used for the experiment was (Fig 1):

## 2.2 Response generation

The open-source Python library of OpenAI is used in this experiment to communicate with GPT-3.5 and generate suitable outputs. The previously crafted prompts were used as the LLM's System Role message to specify its behaviour and input-output structure. As the User Role message, a JSON object was input containing the base station, CUE and D2D pair info. The desired assignment matrix was returned as the Assistant Role message.

**Table 1. Properties of base stations, CUE and D2D pairs.**

|  | Property Name | Description |
|---|---|---|
| **Base Station** | (X,Y) coordinates | This is taken to provide the position of the base station. |
|  | Transmission Power | The transmission power of the base station is given for accurate computations. |
| **Cellular UE** | (X,Y) coordinates | This is taken to provide the position of the Cellular UE. |
|  | Transmission Power | The transmission power of the Cellular UE helps us understand how far the communication can reach and also, consider the noise. |
|  | Target SINR | The target Signal-to-Noise Ratio is taken which must be met by the algorithms. |
| **D2D pairs** | (X,Y) coordinates of Transmitter | This is taken to provide the position of the Transmitter. |
|  | (X,Y) coordinates of Receiver | This is taken to provide the position of the Receiver. |
|  | Transmission Power | The transmission power of the Transmitter is taken in dBm which gives the algorithm an idea about the computations. |
|  | Target SINR | The target Signal-to-Noise Ratio of the Receiver is taken which must be met by the algorithms. |

> You are a telecommunications network optimizer. Think of a 2D space. There's a base station present in (0, 0) coordinate and *Transmission Power* is 46 dBm. This base station communicates with a number of cellular user equipment over a *Carrier Frequency* of 1.7 GHz (LTE) using downlink resources. A number of Device to Device pairs want to communicate by reusing/sharing the existing downlink resources of the cellular user equipment. The number of Cellular UE is greater than the number of D2D pairs. Your goal is to allocate one Cellular UE to one D2D pair so that the *Sumrate* is maximized. To maximize the sumrate you can skip assigning some D2D pairs. You will be provided with a JSON input which will contain *cellInfo object, d2dInfo* object. The cellInfo object will have properties: *totalCells* and an array of cell objects called cells. Each cell object will have properties: *x_coordinate, y_coordinate, transmissionPower, target_SINR*. The d2dInfo object will have properties: *totalD2D* and an array of d2dPair objects called *d2dPairs*. Each d2dPair object will have properties: *receiver_x_coordinate, receiver_y_coordinate, transmitter_x_coordinate, transmitter_y_coordinate, transmitter_transmissionPower, receiver_target_SINR*. All the cell objects and the receiver devices in the d2d pair must obtain their target SNR. Reply to me only in the assignment matrix in JSON format. The output format should be like this: {"assignmentMatrix": [{'cellIndex': number, 'd2dPairIndex': number}]}

**Fig 1. The prompt used for the experiment.**

## 2.3 Algorithm simulation

The performance of LLM is compared with the existing optimal algorithm [39]. In the case of the optimal algorithm, initially, the problem is transformed into a bipartite matching problem with two sets of nodes - one being CUE and the other being D2D pairs. The weight of the edge between a D2D pair and a CUE is their data rate contribution to the total system sumrate while sharing the resources. After the transformation of the problem, the bipartite graph is solved by assigning D2D pairs to the appropriate CUE (assignment of a D2D pair to a CUE means they will share the same resource). Before allocating this edge weight, the system makes sure that sharing doesn't lower the overall data rate and keeps both CUE and D2D pairs' SINR (Signal-to-Interference-plus-Noise Ratio) objectives met. An algorithm following the Hungarian algorithm approach is used to find the optimal assignment of resources between CUE and D2D pairs. This algorithm considers only those assignments that satisfy the constraints, and the calculated data rate is better from a baseline (indicating sharing is feasible without reducing the overall rate).

As shown in Fig 2, the unshared sumrate represents the total sumrate of the system if no D2D pair is sharing resources of any CUE. In essence, it indicates the total system sumrate of CUEs. On the other hand, the Hungarian Graph (not the optimal algorithm) is the solution to the problem following the Hungarian algorithm without considering the situation where an assignment may reduce the overall sumrate.

## 3 Results and discussion

### 3.1 Experimental setup

**3.1.1 Hardware and software environments.** The hardware environments used for the experiment are:

- CPU: i7-1165G7 @ 2.80 GHz
- RAM: 8 GB DDR4 @ 3200 MHz

The software environments used for the experiment are:

- Environment: Jupyter Notebook
- Python Version: 3.11
- C++ Version: 17

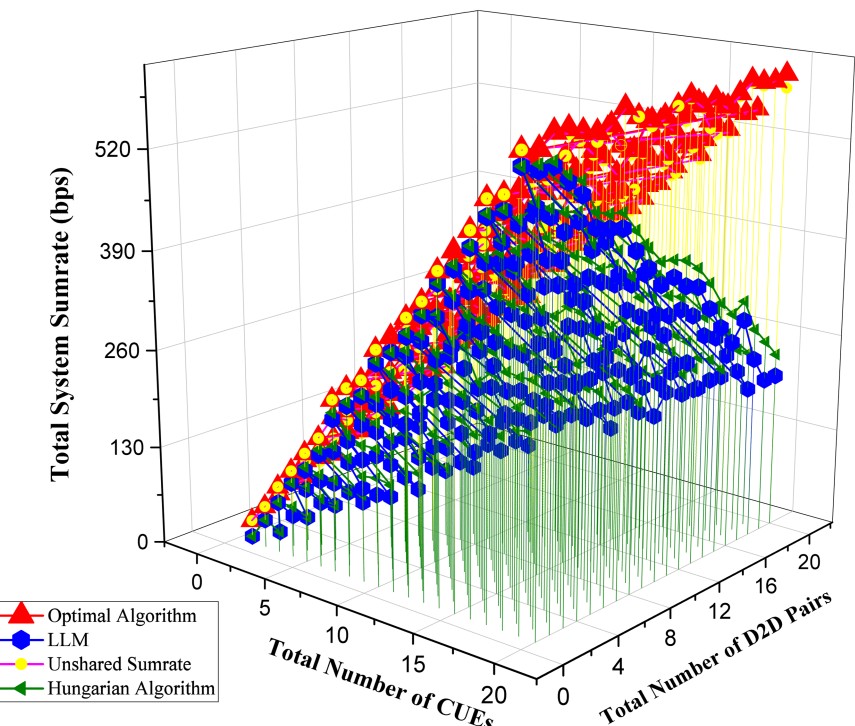

**Fig 2. Comparison of Sumrate between state-of-the-art algorithms and LLM.**

### 3.1.2 Parametric values of the LLMs. The OpenAI chat completion parameters are:

- Model: gpt-3.5-turbo-16k-0613
- Temperature: 0
- Context Window: 16,385 tokens
- Max Output Tokens: 4,096

There are some parametric values necessary for the implementation of the algorithms. To make the outcomes similar, the LLMs are fed those values in the form of a prompt and also as in a JSON file attached to the prompt. The values are as follows (Table 2):

**Table 2. Parametric values provided to the LLMs.**

| Parameter | Value |
| --- | --- |
| Cellular Users | 1 to 20 |
| D2D pairs | 1 to 20 |
| Carrier Frequency | 1.7 GHz |
| Base Station transmission power | 46 dBm |
| Cellular User transmission power | 20 dBm |
| D2D transmission power | 20 dBm |
| $SINR_{c,target}^{DL}$ | Random |
| $SINR_{d,target}^{DL}$ | Random |

## 3.2 Empirical results

For the experiment, the total Cellular UEs are taken from 1 to 20, and the total D2D pairs are also taken from 1 to 20. In order to ensure proper visualization of the sumrates calculated by GPT, we are representing the outputs generated from 5 to 16 of the Cellular UE and D2D pairs rounded to the nearest integer.

Table 3 shows the sumrate calculated for different combinations of Total D2D pairs and Total CUEs. It is evident that LLMs are able to pick the patterns based on the characteristics of the features. To be more precise, LLM could understand that the increase of Total CUE would increase sumrate and increase of Total D2D pairs would decrease the sumrate.

## 3.3 Evaluation metrics

**3.3.1 MSE, RMSE and MAE.** MSE is used to display the proximity of a regression line to a collection of predicted points, which, in this case, is given by LLMs. In order to calculate in the same unit as the response variable and get information on the average divergence between the optimized algorithm and output from LLM, RMSE is used. Along with these metrics, MAE is used to find out the overall disparity between the two observations that were paired, which helps to gauge how accurate continuous variables are.

**3.3.2 Performance ratio.** The ratio between the optimal value and the calculated value is known as the Performance Ratio or Efficiency Ratio. It is a measure of the efficiency of an approach with respect to an optimal approach. In the case of this experiment, the optimal algorithm is the algorithm mentioned in Sect 2.3: Algorithm Simulation and the calculated sumrate is taken as the outputs generated from the LLMs. Now, the Performance Ratio or PR will be:

$$PR = \frac{Sumrate_{LLM}}{Sumrate_{Optimal}} \tag{1}$$

The Average PR is calculated for each value of D2D pairs, keeping the Total Cellular UEs constant and vice versa. Finally, the Mean Average Performance Ratio (MAPR) is the mean of all the calculated Average PRs, which can be written as:

$$MAPR = \frac{1}{m+n} \left( \sum_{i=1}^{m} \frac{\sum_{j=1}^{i} PR(i,j)}{i} + \sum_{j=1}^{n} \frac{\sum_{i=1}^{j} PR(i,j)}{j} \right) \tag{2}$$

**Table 3. Sumrate calculated by GPT for the cell and d2d combination.**

| Total CUE | Total D2D pairs | | | | | | | | | | | |
|---|---|---|---|---|---|---|---|---|---|---|---|---|
|  | 5 | 6 | 7 | 8 | 9 | 10 | 11 | 12 | 13 | 14 | 15 | 16 |
| 5 | 50 | | | | | | | | | | | |
| 6 | 66 | 56 | | | | | | | | | | |
| 7 | 114 | 89 | 65 | | | | | | | | | |
| 8 | 155 | 122 | 101 | 89 | | | | | | | | |
| 9 | 176 | 143 | 133 | 107 | 93 | | | | | | | |
| 10 | 217 | 180 | 159 | 127 | 130 | 115 | | | | | | |
| 11 | 222 | 214 | 194 | 157 | 167 | 134 | 120 | | | | | |
| 12 | 268 | 236 | 231 | 190 | 177 | 165 | 167 | 147 | | | | |
| 13 | 296 | 270 | 232 | 225 | 206 | 194 | 168 | 167 | 160 | | | |
| 14 | 312 | 294 | 284 | 263 | 235 | 229 | 217 | 167 | 173 | 141 | | |
| 15 | 352 | 337 | 303 | 297 | 262 | 235 | 232 | 234 | 180 | 182 | 161 | |
| 16 | 381 | 346 | 299 | 324 | 315 | 268 | 266 | 243 | 227 | 207 | 161 | 188 |

Where,

 m = number of Total Cells considered for that instance

 n = number of Total D2D pairs considered for that instance

 And, PR(i,j) is the Performance ratio of the instance where Total Cells and Total D2D pairs are i and j respectively

## 3.4 Comparison between GPT and simulated algorithms

Some of the widely used algorithms for this task are optimal algorithm obtained from [39], Bipartite Matching and Unshared (sumrate of Cellular UEs without any D2D pairs). The output from optimal produces the most optimal results, so it is taken as ground truth in calculating the errors. If we look at Fig 2, we can visualize that most of the green dots have a greater height than most of the blue dots. This suggests that the Hungarian Graph algorithm produces a higher sumrate than GPT. The optimal algorithm shows that its sumrate remains steady with changes in the variables. The main summary of the figure is that LLMs are able to capture the essence of the task as it is producing results similar to developed algorithms like the Hungarian Algorithm.

**3.4.1 Error calculation.** It is evident that the unshared algorithm produces very similar results to the optimized algorithm from [39]. This is because D2D pairs are not considered in the case. But if it is considered, then there can be large errors. GPT has given an MSE, which is 27% greater than the Bipartite Graph Matching Algorithm, which is one of the algorithms presently in use. It is expected that GPT will not be able to surpass the results produced by these algorithms as it is not fine-tuned to serve this specialized field, and that is exactly what is observed. The purpose of using LLMs like GPT for such a task is to find a sub-optimal result quickly and make fast decisions without going through the hassle of implementation of complex algorithms (Table 4).

**3.4.2 Efficiency calculation.** For calculating the efficiency of the output from GPT with respect to the optimal results, the Performance Ratio is taken as one of the evaluation metrics. The value of the total number of Cellular UEs and the total number of D2D pairs are both taken from 1 to 20 in all possible combinations. For visualization purposes, a portion of the Performance Ratio is shown, along with the Average Performance Ratio and the Mean Average Performance Ratio (MAPR):

In Table 5, the value highlighted in red is the calculated MAPR of GPT with respect to the optimal algorithm. If we want to have a closer look at the continuous progression of the Average PR across each assignment of total CUE, we get: [0.267, 0.432, 0.535, 0.607, 0.592, 0.58, 0.609, 0.638, 0.624, 0.633, 0.644, 0.653, 0.653, 0.647, 0.656, 0.652, 0.649, 0.655, 0.662, 0.656]. This means the LLM performs better when the total CUE increases. Again, if we look at the continuous progression of the Average PR across each assignment of total D2D pairs, we get: [0.874, 0.802, 0.762, 0.719, 0.685, 0.652, 0.614, 0.591, 0.578, 0.541, 0.531, 0.502, 0.496,

**Table 4. Errors from the LLM and simulated algorithms.**

|  | Sumrate | | | Interference | | |
|---|---|---|---|---|---|---|
|  | MSE | RMSE | MAE | MSE | RMSE | MAE |
| GPT | 28920.073 | 170.059 | 143.018 | 5.33E-08 | 0.0002 | 2.10E-05 |
| Bipartite Matching | 22720.694 | 150.734 | 125.675 | 5.30E-08 | 0.0002 | 2.08E-05 |
| Unshared | 22.8214 | 4.777 | 2.805 | 0 | s0 | 4.07E-09 |

**Table 5. MAPR of GPT compared with optimal algorithm.**

| | | Total D2D pairs | | | | | | | Average PR |
|---|---|---|---|---|---|---|---|---|---|
| | | 1 | 2 | 3 | ... | 18 | 19 | 20 | |
| Total CUE | 1 | 0.267 | | | | | | | **0.267** |
| | 2 | 0.666 | 0.198 | | | | | | **0.432** |
| | 3 | 0.755 | 0.487 | 0.364 | | | | | **0.535** |
| | ... | ... | ... | ... | ... | | | | ... |
| | 18 | 0.965 | 0.925 | 0.89 | ... | 0.387 | | | **0.655** |
| | 19 | 0.968 | 0.929 | 0.907 | ... | 0.392 | 0.337 | | **0.662** |
| | 20 | 0.959 | 0.936 | 0.9 | ... | 0.401 | 0.344 | 0.341 | **0.656** |
| **Average PR** | | **0.874** | **0.802** | **0.762** | **...** | **0.393** | **0.341** | **0.341** | **0.58** |

0.455, 0.441, 0.417, 0.422, 0.393, 0.341, 0.341]. This means the LLM performs better when the total D2D pairs are less.

# 4 Result analysis

## 4.1 Performance comparison of the state-of-the-art LLMs

The Bipartite Graph Matching Algorithm shows outstanding sum-rate maximization but is not fully optimized while expanding D2D pairs with a certain total number of Cellular UE. This leads to inferior throughput. A corresponding remark is made regarding the sum-rate results produced by GPT. This demonstrated how well the GPT error calculations match the Bipartite Graph Matching Algorithm's calculations taken into its closest integer. The result suggests that GPT operates in a manner that produces outcomes similar to those of the Bipartite Matching Algorithm by nature.

LLMs such as Llama-2, Palm, etc., are observed to hallucinate heavily for the same prompts used for the GPT experiments. Assignment operations rely heavily on structured data, i.e. JSON. Other LLMs failed to provide structured responses, and thus, the shared assignment matrix could not provide meaningful outcomes.

## 4.2 Efficiency of LLM in sumrate maximization

While maximization can be quite a straightforward task for any AI system, sumrate maximization with several networking constraints is difficult to comprehend. GPT, being the frontrunner in this task, is also struggling to produce praiseworthy results. Compared with the optimized algorithm, GPT has a MAPR of only 0.58. This essentially means that if the total number of cellular users and d2d pairs are between 1 and 20, then we can expect the current GPT model to be effective enough to provide assignments which yield a sumrate which is 58% of the efficient algorithm currently in use at the moment.

## 4.3 Improving LLM for network optimization

The underlying architecture of LLMs can be modified for specific tasks. Fine-tuned LLMs have proved to be more useful compared to generic LLMs when not dealing with diversified tasks [25,30]. In the case of sumrate maximization, if a modified LLM is created, trained on a large corpus with data specific to sumrate calculation and maximization, then our particular task of sumrate maximization would yield better results. But the scope of this manuscript is to focus on the zero-shot capabilities of generic LLMs on the sumrate maximization.

Aside from this, instead of directly using LLMs to provide with resource allocation outputs, LLMs can be used in assisting the user with an output of a subtask. The subtask can

underline specific protocols and assignments which would reduce the search domain for the state-of-the-art algorithms like the Bipartite Matching algorithm.

## 5 Limitations

LLM performance depends mostly on the data used to train it. This is evident in this experiment as GPT, trained with significantly more data, performed much better than other LLMs. Another performance consideration is the context size of the LLMs. GPT is optimized for large contexts, so it hallucinates rarely compared to other LLMs with shorter context sizes. The experiment input prompts had a fairly large number of tokens, which caused other LLMs to hallucinate often. GPT is relatively optimized for structured responses with function calling and JSON mode options, while other models require fine-tuning to produce meaningful structured outputs [40]. This made GPT ideal for this experiment as it could process structured data more reliably than other LLMs which lack JSON data optimizations.

The LLMs are language models by design, and even though they have reasoning capabilities that make them suitable for many unforeseen tasks, these models are still not capable of providing outputs that can compete with existing algorithmic approaches for a couple of reasons. Firstly, LLMs are not designed to run modified algorithmic codes in the background. Secondly, LLMs are non-deterministic by nature, which is not expected while performing such tasks.

## 6 Conclusion and future work

The reasoning capability of LLMs on many tasks has been evaluated and shows satisfactory performance. However, their reasoning ability on complex tasks like sumrate maximization was unexplored. Though they have the ability to produce satisfactory outputs, it is observed that they still cannot compete with state-of-the-art algorithms or approaches. In this experiment, it is seen that the efficiency of LLMs, GPT to be specific, is only around 58%. In cases where numerical and structural information does not play a major role, LLM can produce usable outputs. And the fact that there is no necessity for any prior knowledge makes LLMs a very useful tool. For sumrate maximization, if there are time, knowledge and resource constraints for implementation, LLMs can be an alternative approach for sub-optimal assignments.

One of the major drawbacks of LLMs is they need to be modified to support various tasks. This is one of the improvements that can be integrated into this experiment in the future. With proper fine-tuning, many LLMs which are unusable now can become useful for this specific task [40]. Along with that, changing the desired output structures can sometimes yield a better output.

## Author contributions

**Conceptualization:** Ali Abir Shuvro, Md. Shahriar Islam Bhuiyan, Faisal Hussain, Md. Sakhawat Hossen.

**Data curation:** Md. Shahriar Islam Bhuiyan.

**Formal analysis:** Ali Abir Shuvro, Md. Shahriar Islam Bhuiyan, Faisal Hussain, Md. Sakhawat Hossen.

**Investigation:** Ali Abir Shuvro, Md. Shahriar Islam Bhuiyan, Faisal Hussain, Md. Sakhawat Hossen.

**Methodology:** Ali Abir Shuvro, Faisal Hussain, Md. Sakhawat Hossen.

**Project administration:** Md. Shahriar Islam Bhuiyan, Faisal Hussain, Md. Sakhawat Hossen.

**Resources:** Ali Abir Shuvro, Md. Shahriar Islam Bhuiyan, Faisal Hussain, Md. Sakhawat Hossen.

**Software:** Md. Shahriar Islam Bhuiyan.

**Supervision:** Ali Abir Shuvro, Faisal Hussain, Md. Sakhawat Hossen.

**Validation:** Md. Shahriar Islam Bhuiyan, Md. Sakhawat Hossen.

**Visualization:** Faisal Hussain.

**Writing – original draft:** Ali Abir Shuvro, Md. Shahriar Islam Bhuiyan, Faisal Hussain, Md. Sakhawat Hossen.

**Writing – review & editing:** Ali Abir Shuvro, Faisal Hussain.

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
