## [Decision Letter · Decision Letter 0]

20 May 2024

PONE-D-23-43572Zero-shot Performance Analysis of Large Language Models in Sumrate MaximizationPLOS ONE

Dear Dr. Hossen,

Thank you for submitting your manuscript to PLOS ONE. After careful consideration, we feel that it has merit but does not fully meet PLOS ONE’s publication criteria as it currently stands. Therefore, we invite you to submit a revised version of the manuscript that addresses the points raised during the review process.

We look forward to receiving your revised manuscript.

Kind regards,

Sheetal Kalyani

Academic Editor

PLOS ONE

Journal Requirements:

Additional Editor Comments:

Both the reviewers have some critical concerns. Please address them in detail. More simulations with existing state of art techniques is required. More optimization of the prompt engineering is required. The utility of the work also needs to be discussed in greater detail.

Reviewers' comments:

Reviewer's Responses to Questions

**Comments to the Author**

1. Is the manuscript technically sound, and do the data support the conclusions?

Reviewer #1: Partly

Reviewer #2: No

2. Has the statistical analysis been performed appropriately and rigorously? 

Reviewer #1: No

Reviewer #2: No

3. Have the authors made all data underlying the findings in their manuscript fully available?

Reviewer #1: No

Reviewer #2: Yes

4. Is the manuscript presented in an intelligible fashion and written in standard English?

Reviewer #1: Yes

Reviewer #2: No

5. Review Comments to the Author

Reviewer #1: In this paper, the authors study the performance when leveraging the generative power of LLMs for networking optimization problems. Although the idea is interesting, the reviewer has the following comments:

1. The insights are from the testing of LLM. The paper is more like a technical report of LLM in networking. More insights should be added. For example, how can we improve the LLM to meet the requirement of network optimziation tasks.

2. Moreover, except directly using LLM, is there any other way to use LLM to enhance the network?

Reviewer #2: As the problem is solved using GPT, and the proposed solution is sub-optimal, the reviewer is not sure if the LLM is able to propose a good enough solution. Even though using LLM for sum-rate optimization is unique, but the results are not good enough to ensure that LLM is really useful for all such tasks.

Promt engineering is not explored well for creating algorithms and it needs extensive study and comparison with existing algorithms and show comparable results to establish that LLM can be used for sumrate maximization in wireless communication.

The system model is not present, which make it difficult to follow.

Table 3 does not have explanation.

Fig 2 needs explanation.

6. PLOS authors have the option to publish the peer review history of their article (what does this mean?). If published, this will include your full peer review and any attached files.

Reviewer #1: No

Reviewer #2: No

---

## [Author Response · Author response to Decision Letter 1]

4 Jul 2024

Reviewer 1

Concern 1: The insights are from the testing of LLM. The paper is more like a technical report of LLM in networking. More insights should be added. For example, how can we improve the LLM to meet the requirement of network optimziation tasks.

Response: We appreciate the reviewer's concern. We have added a separate subsection - "Improving LLM for Network Optimization". In this section, we have added our insight on how to improve LLM to meet network optimization task requirements See Page 8, Line 251-258

Concern 2: Moreover, except directly using LLM, is there any other way to use LLM to enhance the network?

Response: In Section 4.3, we proposed an idea about using LLM as an intermediary. It can provide outputs to help other algorithms in achieving better performance in tasks like Sumrate Maximization. See Page 8, Line 259-264.

Reviewer 2

Concern 1: As the problem is solved using GPT, and the proposed solution is sub-optimal, the reviewer is not sure if the LLM is able to propose a good enough solution. Even though using LLM for sum-rate optimization is unique, but the results are not good enough to ensure that LLM is really useful for all such tasks.

Response: We thank the reviewer for pointing this out. LLMs can be used to make quick decisions without having prior algorithmic knowledge. This was unclear earlier and now we have specified it in our manuscript. See Page 7, Line 210-212

Concern 2: Promt engineering is not explored well for creating algorithms and it needs extensive study and comparison with existing algorithms and show comparable results to establish that LLM can be used for sumrate maximization in wireless communication.

Response: Our prompt has evolved through multiple iterations while judging from the outputs of the LLMs. We extensively analyzed the outputs to understand the need of the models and we provided more information to obtain our desired result. We have now added the changes made to each iteration while developing our prompt. See Page 3-4, Line 115-126

Concern 3: The system model is not present, which make it difficult to follow.

Response: Section 2 - Methodology has subsections which are the steps taken by us for conducting our experiment. This is now specified in the manuscript. See Page 3, Line 104-105

Concern 4: Table 3 does not have explanation.

Response: We are grateful to the reviewer for identifying this as it will definitely help improve the quality of the manuscript. We have added a proper explanation to the table to assist the readers. See Page 5, Line 166-170

Concern 5: Fig 2 needs explanation.

Response: A proper explanation of the figure with all the features is currently added. A summary of the main gist of the figure is also added in one sentence. See Page 7, Line 199-202

---

## [Decision Letter · Decision Letter 1]

12 May 2025

PONE-D-23-43572R1Zero-shot Performance Analysis of Large Language Models in Sumrate MaximizationPLOS ONE

Dear Dr. Hossen,

Thank you for submitting your manuscript to PLOS ONE. After careful consideration, we feel that it has merit but does not fully meet PLOS ONE’s publication criteria as it currently stands. Therefore, we invite you to submit a revised version of the manuscript that addresses the points raised during the review process.

We look forward to receiving your revised manuscript.

Kind regards,

Divya Chaudhary, Ph.D.

Academic Editor

PLOS ONE

Journal Requirements:

Additional Editor Comments:

Minor Revision

Both the reviewer comments should be addressed for acceptance of the paper.

Reviewer 1

Most my previous comments are addressed. This paper can provide some insights for the applications of LLM in the network issues. However, one important things is that most refs discussed in this paper is about the LLM from the AI perspective. Considering that recently there are many works that use LLM in the network optimization, e.g., together with reinforcement learning or RAG, it is necessary to discuss these papers to provide the readers a detailed background.

Reviewer 2

To strengthen your manuscript’s reproducibility and clarity, please consider the following revisions:

• Specify the hardware and software environment used for the experiment

• Report the average turnaround time from prompt submission to GPT response when using your JSON input

• It would be helpful if the authors provided insight into the parameters used for the OpenAI ChatCompletion to generate the response, e.g., max tokens, conversation: system, user, etc.

• Replace the generic phrase “In the result comparison graph” used in section 2.3 with an explicit pointer (e.g., “As shown in Figure 3…”). To ensure the graph refrenced is understood by the readers.

Reviewers' comments:

Reviewer's Responses to Questions

**Comments to the Author**

1. If the authors have adequately addressed your comments raised in a previous round of review and you feel that this manuscript is now acceptable for publication, you may indicate that here to bypass the “Comments to the Author” section, enter your conflict of interest statement in the “Confidential to Editor” section, and submit your "Accept" recommendation.

Reviewer #1: (No Response)

Reviewer #3: (No Response)

2. Is the manuscript technically sound, and do the data support the conclusions?

Reviewer #1: (No Response)

Reviewer #3: Partly

3. Has the statistical analysis been performed appropriately and rigorously? 

Reviewer #1: (No Response)

Reviewer #3: N/A

4. Have the authors made all data underlying the findings in their manuscript fully available?

Reviewer #1: (No Response)

Reviewer #3: Yes

5. Is the manuscript presented in an intelligible fashion and written in standard English?

Reviewer #1: (No Response)

Reviewer #3: Yes

6. Review Comments to the Author

Reviewer #1: Most my previous comments are addressed. This paper can provide some insights for the applications of LLM in the network issues. However, one important things is that most refs discussed in this paper is about the LLM from the AI perspective. Considering that recently there are many works that use LLM in the network optimization, e.g., together with reinforcement learning or RAG, it is necessary to discuss these papers to provide the readers a detailed background.

Reviewer #3: To strengthen your manuscript’s reproducibility and clarity, please consider the following revisions:

• Specify the hardware and software environment used for the experiment

• Report the average turnaround time from prompt submission to GPT response when using your JSON input

• It would be helpful if the authors provided insight into the parameters used for the OpenAI ChatCompletion to generate the response, e.g., max tokens, conversation: system, user, etc.

• Replace the generic phrase “In the result comparison graph” used in section 2.3 with an explicit pointer (e.g., “As shown in Figure 3…”). To ensure the graph refrenced is understood by the readers.

7. PLOS authors have the option to publish the peer review history of their article (what does this mean?). If published, this will include your full peer review and any attached files.

Reviewer #1: No

Reviewer #3: No

---

## [Author Response · Author response to Decision Letter 2]

12 Jun 2025

Reviewer 1

Most my previous comments are addressed. This paper can provide some insights for the applications of LLM in the network issues. However, one important things is that most refs discussed in this paper is about the LLM from the AI perspective. Considering that recently there are many works that use LLM in the network optimization, e.g., together with reinforcement learning or RAG, it is necessary to discuss these papers to provide the readers a detailed background.

Response: We appreciate the reviewer's concern. We have discussed another manuscript in our Related Works section which is focused on this particular suggestion.

Reviewer 2

To strengthen your manuscript’s reproducibility and clarity, please consider the following revisions:

• Specify the hardware and software environment used for the experiment

Response: We have added another subsection (Section 3.1.1) titled "Hardware and Software Environments" under the Section Experimental Setup where we discussed every experimental environments used.

• It would be helpful if the authors provided insight into the parameters used for the OpenAI ChatCompletion to generate the response, e.g., max tokens, conversation: system, user, etc.

Response: We thank the reviewer for pointing this out. We have now added another subsection, "Parametric Values of the LLMs," where we discuss these in detail for the readers.

• Replace the generic phrase “In the result comparison graph” used in section 2.3 with an explicit pointer (e.g., “As shown in Figure 3…”). To ensure the graph refrenced is understood by the readers.

Response: We corrected this cosmetic issue and accepted the reviewers suggestion.

---

## [Decision Letter · Decision Letter 2]

4 Jul 2025

PONE-D-23-43572R2Zero-shot Performance Analysis of Large Language Models in Sumrate MaximizationPLOS ONE

Dear Dr. Hossen,

Thank you for submitting your manuscript to PLOS ONE. After careful consideration, we feel that it has merit but does not fully meet PLOS ONE’s publication criteria as it currently stands. Therefore, we invite you to submit a revised version of the manuscript that addresses the points raised during the review process.

We look forward to receiving your revised manuscript.

Kind regards,

Divya Chaudhary, Ph.D.

Academic Editor

PLOS ONE

Journal Requirements:

**Additional Editor Comments:**

Please include the reviewer comments in the submission.

Reviewers' comments:

Reviewer's Responses to Questions

**Comments to the Author**

1. If the authors have adequately addressed your comments raised in a previous round of review and you feel that this manuscript is now acceptable for publication, you may indicate that here to bypass the “Comments to the Author” section, enter your conflict of interest statement in the “Confidential to Editor” section, and submit your "Accept" recommendation.

Reviewer #3: All comments have been addressed

Reviewer #4: All comments have been addressed

2. Is the manuscript technically sound, and do the data support the conclusions?

Reviewer #3: Partly

Reviewer #4: Yes

3. Has the statistical analysis been performed appropriately and rigorously? 

Reviewer #3: N/A

Reviewer #4: Yes

4. Have the authors made all data underlying the findings in their manuscript fully available?

Reviewer #3: (No Response)

Reviewer #4: Yes

5. Is the manuscript presented in an intelligible fashion and written in standard English?

Reviewer #3: Yes

Reviewer #4: Yes

6. Review Comments to the Author

Reviewer #3: I commend the authors for their efforts in improving the manuscript. However, I suggest a few suggestions to further strengthen the manuscript:

1. Although the introduction is brief and to the point, it would be beneficial to include a short explanation of what sum-rate maximization entails, particularly for readers who may not be specialists in this area. This addition would enhance accessibility for a broader audience.

2. The manuscript stated a valid concerns regarding the limitations of large language models (LLMs) in reasoning in the related works section (line 59 – 63) of the manusctipt. However, recent advancements have introduced techniques such as chain-of-thought prompting, and scratchpad reasoning which have significantly enhanced the performance of LLMs in structured reasoning tasks. Instead of broadly stating that LLMs struggle with reasoning, I recommend refining this point to reflect that while current models have improved, their reasoning abilities can still be fragile or dependent on context.

3. Additionally, the manuscript employs GPT-3.5-Turbo, which is now considered a relatively outdated version of OpenAI's models. It would be helpful to explain the rationale behind selecting this version. If possible, consider rerunning the experiments using more recent models with stronger reasoning capabilities, such as GPT-4o or o3-mini, as they may yield improved results and strengthen your conclusions.

Reviewer #4: The authors have adequately addressed all the comments raised in a previous round of review.

# # # #

7. PLOS authors have the option to publish the peer review history of their article (what does this mean?). If published, this will include your full peer review and any attached files.

Reviewer #3: No

Reviewer #4: No

---

## [Author Response · Author response to Decision Letter 3]

7 Jul 2025

1. We appreciate the reviewer's concern. We have discussed about sumrate maximization in our introduction for the readers to have a clearer idea about the topic.

2. We have discussed about chain-of-thought prompting and scratchpad reasoning in our manuscript taking into account the reviewer's suggestion. We thank the reviewers for this valuable feedback.

3. We thank the reviewer for pointing this out. The scope of the work, when submitted, was with GPT 3.0 and we are certain that newer models might outperform this model. But the research is regarding the then version of GPT and if that shows significant performance boost (which it does), we can conclude that newer models would follow the trend.

---

## [Decision Letter · Decision Letter 3]

21 Jul 2025

Zero-shot Performance Analysis of Large Language Models in Sumrate Maximization

PONE-D-23-43572R3

Dear Md Sakhawat Hossen,

We’re pleased to inform you that your manuscript has been judged scientifically suitable for publication and will be formally accepted for publication once it meets all outstanding technical requirements.

Kind regards,

Divya Chaudhary, Ph.D.

Academic Editor

PLOS ONE

Additional Editor Comments (optional):

Reviewers' comments:

Reviewer's Responses to Questions

**Comments to the Author**

1. If the authors have adequately addressed your comments raised in a previous round of review and you feel that this manuscript is now acceptable for publication, you may indicate that here to bypass the “Comments to the Author” section, enter your conflict of interest statement in the “Confidential to Editor” section, and submit your "Accept" recommendation.

Reviewer #3: All comments have been addressed

2. Is the manuscript technically sound, and do the data support the conclusions?

Reviewer #3: Partly

3. Has the statistical analysis been performed appropriately and rigorously? 

Reviewer #3: N/A

4. Have the authors made all data underlying the findings in their manuscript fully available?

Reviewer #3: Yes

5. Is the manuscript presented in an intelligible fashion and written in standard English?

Reviewer #3: Yes

6. Review Comments to the Author

Reviewer #3: (No Response)

7. PLOS authors have the option to publish the peer review history of their article (what does this mean?). If published, this will include your full peer review and any attached files.

Reviewer #3: No
